# Fecal Microbiota Transplantation Controls Murine Chronic Intestinal Inflammation by Modulating Immune Cell Functions and Gut Microbiota Composition

**DOI:** 10.3390/cells8060517

**Published:** 2019-05-28

**Authors:** Claudia Burrello, Maria Rita Giuffrè, Angeli Dominique Macandog, Angelica Diaz-Basabe, Fulvia Milena Cribiù, Gianluca Lopez, Francesca Borgo, Luigi Nezi, Flavio Caprioli, Maurizio Vecchi, Federica Facciotti

**Affiliations:** 1Department of Experimental Oncology, IEO European Institute of Oncology IRCCS, 20139 Milan, Italy; claudia.burrello@ieo.it (C.B.); angelidominique.macandog@ieo.it (A.D.M.); angelicajulieth.diazbasabe@ieo.it (A.D.-B.); francesca.borgo@ieo.it (F.B.); luigi.nezi@ieo.it (L.N.); 2Department of Pathophysiology and Transplantation, Università degli Studi di Milano, 20135 Milan, Italy; mariarita.giuffre’@ieo.it (M.R.G.); Flavio.caprioli@gmail.com (F.C.); Maurizio.vecchi@unimi.it (M.V.); 3Department of Oncology and Hemato-oncology, Università degli Studi di Milano, 20135 Milan, Italy; 4Pathology Unit, Fondazione IRCCS Cà Granda, Ospedale Maggiore Policlinico, 20135 Milan, Italy; Fulviamilena.cribiu@policlinico.mi.it (F.M.C.); gianluca.lopez@policlinico.mi.it (G.L.); 5Gastroenterology and Endoscopy Unit, Fondazione IRCCS Cà Granda, Ospedale Maggiore Policlinico, 20135 Milan, Italy

**Keywords:** FMT, IBD, T cells, microbiota

## Abstract

Different gastrointestinal disorders, including inflammatory bowel diseases (IBD), have been linked to alterations of the gut microbiota composition, namely dysbiosis. Fecal microbiota transplantation (FMT) is considered an encouraging therapeutic approach for ulcerative colitis patients, mostly as a consequence of normobiosis restoration. We recently showed that therapeutic effects of FMT during acute experimental colitis are linked to functional modulation of the mucosal immune system and of the gut microbiota composition. Here we analysed the effects of therapeutic FMT administration during chronic experimental colitis, a condition more similar to that of IBD patients, on immune-mediated mucosal inflammatory pathways. Mucus and feces from normobiotic donors were orally administered to mice with established chronic Dextran Sodium Sulphate (DSS)-induced colitis. Immunophenotypes and functions of infiltrating colonic immune cells were evaluated by cytofluorimetric analysis. Compositional differences in the intestinal microbiome were analyzed by 16S rRNA sequencing. Therapeutic FMT in mice undergoing chronic intestinal inflammation was capable to decrease colonic inflammation by modulating the expression of pro-inflammatory genes, antimicrobial peptides, and mucins. Innate and adaptive mucosal immune cells manifested a reduced pro-inflammatory profile in FMT-treated mice. Finally, restoration of a normobiotic core ecology contributed to the resolution of inflammation. Thus, FMT is capable of controlling chronic intestinal experimental colitis by inducing a concerted activation of anti-inflammatory immune pathways, mechanistically supporting the positive results of FMT treatment reported in ulcerative colitis patients.

## 1. Introduction

The intestinal mucosa is a complex environment where the host and the gut microbiome establish a mutualistic relationship [1]. Maintenance of intestinal homeostasis relies on different mechanisms aimed at physically separating the commensal microbial ecology from the intestinal lamina propria, controlling colonization by pathogenic bacteria, and preserving the mucosal immune cell functions [1]. The intrinsic resilience of the system allows subtle alterations of this equilibrium [2], unless chronic inflammatory insults or underlying genetic alterations establish a stable diversion from the normobiotic microbiome composition, as it is reported in patients suffering from inflammatory bowel diseases (IBD, Crohn’s disease (CD), and Ulcerative colitis (UC)) [3,4].

IBD patients and their first-degree relatives manifest a reduced biodiversity in both mucus-associated and fecal microbial communities, including bacteria, fungi, and viruses [3,5,6]. Expansion of pro-inflammatory bacteria, such as *Enterobacteriaceae* and *Fusobacteriaceae*, and depletion of phyla with anti-inflammatory capabilities, such as *Firmicutes*, are observed in IBD patients [7]. Intestinal dysbiosis in IBD patients seem to temporally precede intestinal inflammation, as it has been observed in the ileal mucosa of children with treatment-naïve IBD [3]. These observations are mechanistically supported by the capability of dysbiotic microbiota or single commensal bacterial species to induce intestinal inflammation when transferred in germ-free mice [8].

Consistently, the chronic maintenance of a dysbiotic microbiota seems to be linked to some of the genetic alterations reported in IBD patients, e.g., *NOD* genes encoding for proteins and receptors involved in bacterial sensing [9]. Moreover, the chronic damage of the intestinal epithelium observed in IBD patients is a key event favoring bacterial translocation into the lamina propria, thus promoting the recognition of antigens derived from the dysbiotic microflora by pathogenic T cells [10].

Altogether these events support the hypothesis that an altered luminal microbiota composition fuels the prolonged recruitment and activation of immune cell types into the intestinal mucosa of IBD patients, contributing to chronically maintaining intestinal inflammation [10].

Both innate and adaptive mucosal immune cells are involved in pathogenic activities during intestinal inflammation [11]. Conventional CD4+ T helper (Th) cells have been shown to play key role in coordinating intestinal inflammatory responses [12,13], although other mucosal T cell populations. including Type 1 [14] and Type 2 Natural Killer T (NKT) cells [15,16], contribute to sustain intestinal inflammation. Conversely, regulatory T cell populations, such as Treg [17] and Tr1 cells [18], participate in the control of Th17-mediated pathogenic responses in IBD patients.

Therefore, besides the control of the disease, restoration of a normobiotic core intestinal ecology [14,19,20] is becoming a long-term therapeutic goal in various gastrointestinal disorders [21]. One widely explored option to manipulate and treat disorders associated with microbiota alterations is fecal microbiota transplantation (FMT), i.e., the infusion of a healthy donor feces in the intestine of an affected recipient. At present FMT is receiving positive consideration thanks to therapeutic efficacy against refractory and recurrent *Clostridium difficile* infections (CDI) [22,23], but promising evidence is also accumulating from ongoing randomized clinical trials (RCTs) involving mild-to-moderate UC patients [24,25,26].

Nonetheless, at present it is only partially understood how FMT functionally modulates the intestinal immune system [27,28]. We have recently shown that therapeutic FMT administered during acute experimental intestinal inflammation directly modulates both innate and adaptive mucosal immune responses towards control of intestinal inflammation [19]. Specifically, we showed that FMT is not only capable of restoring normobiosis and metabolic functions associated with beneficial commensals, but it also simultaneously activates multiple anti-inflammatory mechanisms targeting mucosal immune cell types and effector pathways linked to the production of tolerogenic IL10 by mucosal innate and adaptive cell subsets.

Here, we extended our analyses to a chronic setting of intestinal inflammation. Our goal is to understand if a similar regulatory axis involving microbial normobiosis restoration and immune system regulation occurs when FMT is administered to chronically inflamed recipients, thus shedding light on the beneficial mechanism of action of FMT in IBD patients.

## 2. Materials and Methods 

### 2.1. Mice

8–10 weeks old C57BL/6 (Charles River, IT) and CXCR6^EGFP/+^ (B6.129P2-*Cxcr6tm1Litt*/J; IMSR_JAX: 005693, on C57BL/6 background, JAX, USA) female mice were housed at the European Institute of Oncology (IEO) animal facility in Specific pathogen free (SPF) conditions. Experimental groups of mice receiving different FMT treatments were separated in distinct cages and randomly assigned to the different experimental groups.

Animal procedures were approved by Italy’s Ministry of Health (Auth. 415/2017).

### 2.2. Experimental Colitis Models

For the induction of chronic intestinal colitis, mice were given 2% (*w*/*v*) dextran sodium sulphate (DSS, MW 40 kD; TdB Consultancy) in their drinking water for 3 cycles of 7 days of treatment, followed by 14 days of water recovery. Mice were not treated with analgesics. Mice were weighted twice a week to determine the weight curve. At the end of the treatment mice were sacrificed and colons were collected. Firstly, colons were measured to determine their length, then they were divided into portions for different analyses, i.e. fixed in 10% formalin for histology, or in paraformaldehyde, L-Lysine pH 7.4 and NaIO4 (PLP buffer) for immunofluorescence, or snap-frozen for RNA and protein extraction and for lamina propria mononuclear cells (LPMC) isolation.

Experimental groups were composed by *n* = 8 untreated, *n* = 10 chronic DSS (cDSS)- treated, *n* = 10 chronic DSS + Faecal Microbiota Transplantation (cDSS + FMT) treated mice in 2 independent experiments.

### 2.3. Fecal Microbiota Transplantation (FMT)

To perform FMT treatment, mucus (first day) and faeces (second, third and fourth days) preparations from untreated (normobiotic) donors were orally administered to colitic recipient mice. The first day mucus was scraped from colons of untreated mice, diluted in Phosphate Buffered saline (PBS) and dispensed to recipients at 1:1 ratio by oral gavage. Next (days two, three and four of FMT treatment), faeces were collected from colons of untreated mice, diluted in PBS (50 mg/mL) and orally dispensed to recipients (10 mg/mouse) five days after the beginning of each DSS cycle.

### 2.4. Histological Analysis

Colonic tissues were processed with a LEICA PELORIS processor and then embedded in paraffin. Inclusion of murine tissue samples was performed with a robotic system (SAKURA Tissue-Tek) as previously described [29,30]. Hematoxylin and Eosin staining was performed as described in [29,30] and snapshots of tissue histology were taken using an Aperio CS2 microscope (scanning resolution 50,000 pixels per inch, 0.5 µm per pixel with 10× objective and 2.5 µm per pixel when scanning at 4×). Disease activity scoring was performed according to the evaluation of the criteria listed in Appendix A.

### 2.5. Quantitative Reverse Transcription PCR (RT-qPCR) of Tissue mRNA

Total RNA was isolated from murine colonic tissues by using TRIZOL and Quick-RNA MiniPrep (ZymoResearch) following the manufacturer’s specifications and the *MetaHIT* project guidelines. 1 µg of total RNA was used to generate cDNAs by utilizing the reverse transcription kit (Promega). qPCR were performed to evaluate gene expression levels, normalized to the expression of the *Rpl32* gene. The primer sequences are listed in Appendix A.

### 2.6. Tissue ELISA of Murine IL-10

Portions of colonic tissues were homogenized in 300 µL RIPA Buffer (Cell Signaling Technology) containing Phosphatase (Sigma) and Protease (Complete Ultra tablets, Roche) inhibitors, incubated under slow rotation (4 °C for 30 min) and then centrifuged at 13,000 rpm (16.2 g) for 15 min at 4 °C. Quantification of the supernatants was performed using Bradford Assay (BioRad) and measured with NanoDrop. 6.25 µg of lysate were used to determine tissue mIL-10 using the ELISA assay (Purified anti-mouse IL-10 and Biotin anti-mouse IL-10, Biolegend) according to the manufacturer’s instructions.

### 2.7. Murine Cell Isolation

To isolate lamina propria mononuclear cells (LPMC), Peyer’s Patches were firstly removed and then colonic tissue was incubated with 5 mM EDTA at 37 °C for 30 min, followed by digestion with collagenase IV and DNase (37 °C for 1 h). Colonic lamina propria lymphocytes (LPL) were then separated with a Percoll gradient.

In some experiments, cytokine secretion by isolated LPL was determined upon re-stimulation in vitro for 3 h with PMA/Ionomycin in the presence of Brefeldin A.

### 2.8. Flow Cytometry Analysis

For immunephenotyping, cells were incubated with combinations of directly conjugated antibodies (annotated in Appendix A). iNKT cells were identified either by CXCR6-^EGFP^ expression or by mCD1d:PBS57 Tetramer (NIH Tetramer core facility) staining.

To detect intracellular cytokines, restimulated cells were fixed and permeabilized with Cytofix/Cytoperm (BD) before the addition of antibodies. Samples were passed on a FACSCanto II flow cytometer (BD). Data were analysed using FlowJo software (BD) and nonviable cells were excluded by gating.

### 2.9. Immunofluorescence

Colonic samples were fixed overnight in paraformaldehyde, L-Lysine pH 7.4 and NaIO4 (Periodate-Lysine-Paraformaldeyde (PLP) buffer, freshly prepared in house). Samples were washed, dehydrated in 20% sucrose (at least 4 h) and included in the optimal cutting temperature (OCT) cryo-embedding medium (Sakura, Alpen aan den Rijin, The Netherlands). Sections (10μm-thick) were re-hydrated with 0.1M Tris HCl pH 7.4 buffer, blocked with 0.3% triton X-100, Tris-HCl buffer 0.1 M containing 2% FBS. Slides were incubated with the primary antibody (anti-ZO-1 FITC, 1:100) for 2 h. Nuclei were counterstained with 4′,6-diamidino-2-phenylindole (DAPI) (1:30,000 Roche Biochemicals, Monza, IT) and mounted with Vectashield (Vectorlabs, Burlingame, CA).

### 2.10. Bacterial DNA Extraction and 16S rRNA Gene Sequencing

Feces and mucus scraped from the colon were stored at −80 °C until the DNA was extracted with G NOME DNA isolation kit (MP Biomedicals) following the manufacturer’s instructions. 

The 16S rRNA gene amplicon libraries were performed with a two-step barcoding approach according to Borgo et al. [31]. Briefly, the 16S rRNA gene was initially amplified by interest-specific primers targeting V3 (5′-TCGTCGGCAGCGTCAGATGTGTATAAGAGACAGCCTACGGGNGGCWGCAG-3′), 

V4 (5′-TCGTCGGCAGCGTCAGATGTGTATAAGAGACAGCCTACGGGNGGCWGCAG-3′) regions coupled with overhang adapters. The reaction was carried out in 25 µL volume containing 6 ng/µL microbial DNA, 1 µM of each primer, and 2x KAPA HiFi HotStart ReadyMix (Roche, Italy). 

The following PCR program was used: initial denaturation at 95 °C for 3 min, followed by 25 cycles consisting of denaturation (95 °C for 30 s), annealing (55 °C for 30 s) and extension (72 °C for 30 s), and a final extension step at 72 °C for 5 min. PCR products were analyzed by 1% agarose gel electrophoresis for quantity and quality. Expected product size after the Amplicon PCR step is about 550 bp.

DNA samples resulting from the PCR step were amplified with dual-index primers using Nextera Xt Index Kit V2 Set A (Illumina, San Diego, CA). Each sample possessed specific barcode sequences at the 5′- and 3′-end of the PCR amplicon for sample identification in the pooled library.

Library concentration and exact product size were measured using the Agilent 2100 Bioanalyzer System (Agilent, Santa Clara, CA, USA). A 20 nM pooled library and a PhiX control v3 (20 nM) (Illumina, San Diego, CA) were mixed with 0.2 N fresh NaOH and hybridization buffer HT1 (Illumina) to produce a final concentration at 10 pM each. The resulting library was mixed with the PhiX control v3 (5%, *v*/*v*) (Illumina) and injected on a Miseq Reagent Nano Kit V2 500 Cycles to obtain a paired-end 2 × 150 bp sequencing. Sequencing was performed at the IEO Genomic Unit.

### 2.11. Statistical Analysis

#### 2.11.1. Data Analysis

Statistical significance was calculated with unpaired two-tailed Mann-Whitney test (two-tailed) and expressed as mean ± SEM. Values of *p* < 0.05 (*), *p* < 0.01 (**), and *p* < 0.001 (***) were considered as statistically significant. Outliers were detected with Grubb’s test.

#### 2.11.2. Microbiota Sequence Analysis

The 250 bp 16S reads were processed through QIIME2 (version 2018.11) [32] as follows: (1) Following visualization of demultiplexed samples and the average quality across the reads, quality filtering, dereplicating, and chimera filtering were performed using the DADA2 [33] plugin within QIIME2, with the following parameters: –p-trunc-len-f 250, –p-trunc-len-r 242, –p-trim-left-f 0, –p-trim-left-r 0, –p-trunc-q 2, and using consensus as the chimera filtering method; (2) a phylogenetic tree was generated for downstream core diversity analyses using Multiple Alignment using Fast Fourier Transform (MAFFT) [34] for multi-sequence alignment and IQ-TREE [35] ultrafast bootstrap for tree generation, with the following parameters: –p-perturb-nni-strength 0.2 and –p-stop-iter 200; (3) alpha and beta diversity core metrics were assessed using the qiime diversity command; (4) taxonomy classification was performed using the qiime feature-classifier classify-sklearn feature, using a Naïve Bayes classifier [36] trained on SILVA database (release 132, 99% operational Taxonimic Units (OTUs) full-length 16S rRNA sequences), available from the QIIME2 website (https://docs.qiime2.org/2018.11/data-resources/). Taxonomy count tables for phylum, family, and species were generated after taxonomy classification and exported for manipulation and plotting outside of QIIME2.

The QIIME2 frequency table, taxonomy table, phylogenetic tree, and metadata were imported as “physeq” R objects (R Core Team, 2013) using the phyloseq package [37]. Principal coordinate analysis of unweighted UniFrac beta-diversity distance matrix exported from QIIME2 and alpha diversity measurements of physeq object were determined by microbiomeSeq, with alpha-diversity pairwise comparisons assessed by ANOVA. Beta-dispersion pairwise comparisons on unweighted UniFrac values was performed in R using betadisper in vegan [38]. Pairwise PERMANOVA of unweighted UniFrac values was performed separately using pairwise Adonis in R using the default Bonferonni *p*-value correction (https://github.com/pmartinezarbizu/pairwiseAdonis). Comparison of abundances at different taxonomic levels was visualized with bar plots using ggplot2 using the exported taxonomy tables from QIIME2, and Kruskal-Wallis for significance testing. Differential abundance at genus level was performed on physeq object using DESeq2, as implemented within microbiomeSeq, with modifications on the differential_abundance and plot_signif scripts to allow log2 fold change comparisons between chronically DSS-treated (cDSS) and chronically DSS-treated with FMT treatment (cDSS + FMT) groups. Statistical significance of log2 fold changes was assessed using the default Wald test with Benjamin-Hochberg *p*-value correction in DESeq2. Cut-off for all significance tests was set at *p* < 0.05.

#### 2.11.3. PICRUSt Analysis

Phylogenetic Investigation of Communities by Reconstruction of Unobserved States (PICRUSt) was used to predict metagenome function from the 16S rRNA gene data. QIIME2 output was made compatible with PICRUSt by performing closed-reference OTU picking with qiime vsearch, using Greengenes database (13_8 99% OTUs release). The output feature table was then exported into Biological Observation Matrix (BIOM) format and run through the PICRUSt v1.1.3 workflow to normalize the copy number, predict metagenomes, and categorize by function using the Kyoto Encyclopedia of Genes and Genomes (KEGG) database for metabolic pathways. The biom_to_stamp.py script from the Microbiome Helper [39] repository was used to convert the output BIOM file from PICRUSt into Statistical Analysis of metagenomic profiles (STAMP)-compatible spf format. Bray–Curtis distances were used to determine the similarity of samples based on metagenomic composition. Differences in the taxa and predicted molecular functions were analyzed by the linear discriminant analysis (LDA) effect size (LEfSe) with default settings (alpha value for the factorial Kruskal–Wallis test among classes = 0.05; threshold on the logarithmic LDA score for discriminative features = 1.5).

## 3. Results

Therapeutic FMT reduces signs of inflammation in experimental chronic colitis. 

To evaluate if FMT might exert beneficial effects in a chronic intestinal pathologic condition similar to that observed in human IBD patients, experimental intestinal inflammation was induced in mice by chronic administration of Dextran Sodium Sulphate (DSS, Figure 1). Mucus and feces derived from normobiotic mice were repeatedly transferred into colitic mice by oral gavage 5 days after the initiation of each DSS treatment, according to the scheme described in Figure 1A. During the course of the treatment the body weight of FMT-treated mice did not differ from that of cDSS- treated mice (Figure 1B). The histological score was also similar among colitic FMT-treated and untreated mice (Figure 1C). Conversely, at sacrifice FMT-treated mice showed a mild increase in colonic length (Figure 1D). Moreover, in accordance to what was observed when FMT was administered during acute colitis [19], the normobiotic FMT treatment reduced intestinal inflammation, as indicated by a strong decrease in the colonic expression of the pro-inflammatory genes interferon gamma (*Ifng*), tumor necrosis factor (*Tnf*), interleukins 1*β* (*Il1b*), *17* (*Il17*), and 6 (*Il6*) (Figure 1E).

The intestinal barrier contributes to gut homeostasis through a well-structured epithelial layer, the secretion of antimicrobial peptides (AMPs), and the formation of a multi-layer mucous barrier [40]. The architecture of the tight junction protein ZO-1, affected by chronic DSS treatment, was restored to normal levels upon normobiotic FMT administration (Figure 2A). In agreement with previous results [41], expression of mucin genes, such as *Muc1*, *Muc3*, and *Muc4*, and of antimicrobial peptide genes (*Camp* and *S100a8*) were downregulated in FMT-treated mice (Figure 2B).

Thus, our results show that therapeutic FMT ameliorates intestinal inflammation in settings that more closely resemble human IBD pathology, such as that of experimental chronic colitis. 

### 3.1. Therapeutic FMT Influences the Activation Status of Colonic T Cell Populations

Gut-associated immune cell populations critically contribute to initiate and sustain intestinal inflammation, both in IBD patients [11] and in experimental models of intestinal inflammation [42]. Thus, we asked if therapeutic FMT administration during chronic colitis might induce variations in the frequency and phenotype of the colonic immune infiltrate.

At sacrifice, we observed neither frequency nor absolute numbers variations in Invariant Natural Killer T (iNKT) and conventional CD4+ and CD8+ T cells (Figure 3A and Appendix A) in mice treated or not with FMT. Nonetheless, T cell populations isolated from colons of FMT-treated mice showed a reduced proliferative capacity as compared to those isolated from colitic mice that had not received the FMT treatment (Figure 3B–D). Phenotypically, mice receiving FMT displayed a lower proportion of CD4^+^ T and CD8+ T cells expressing the cytotoxicity-associated molecule CD107a [43], additionally supporting the observation that colonic T cells in FMT-treated mice manifest a reduced pro-inflammatory phenotype. To note, at sacrifice CD69 expression on T cells was similar in FMT-treated and untreated mice (Appendix A).

Interestingly, the expansion of the innate lymphocytes ILC2 and ILC3, but not of ILC1, observed in the lamina propria of chronically DSS-treated mice, was reduced upon FMT treatment (Appendix A).

At sacrifice, similar to what was observed in T cell populations, also the frequency (Figure 4A) and the absolute numbers (Figure 4B) of professional antigen-presenting cells (APCs), including dendritic cells, macrophages, B cells, and neutrophils, did not substantially differ between FMT-treated and not-FMT-treated groups. On the contrary, the numbers of colonic MHC-II-expressing professional APCs were reduced, albeit not significantly, in FMT-treated mice (Figure 4C), suggesting a functional modulation of mucosal APCs’ functions by FMT.

Taken together, these data indicate a specific effect of therapeutic FMT on the functional status of infiltrating innate and adaptive immune cell populations.

### 3.2. Therapeutic FMT Targets the Cytokine Profile of Colonic Immune Cell Populations

Cytokines produced by mucosal immune cells play a pivotal role in the maintenance of homeostasis, as in the initiation and propagation of intestinal inflammation [44].

The colonic protein levels of the homeostatic cytokine IL-10 were significantly reduced in colitic mice and restored to those of healthy mice upon FMT treatment (Figure 5A).

Colonic T cells and Lineage (CD11c, CD11b, and CD19) positive leukocytes isolated from chronically inflamed colitic mice showed an increase in the secretion of pro-inflammatory cytokines, including interferon gamma (IFNγ) and interleukins 17 (IL-17) and 13 (IL-13), which was significantly reduced upon FMT treatment (Figure 5B), indicating that the observed amelioration of the inflammatory conditions of FMT-treated mice (Figure 1) was associated not only with a reduced proliferative capacity (Figure 2) but also with an overall reduced pro-inflammatory potential of mucosal immune cells.

### 3.3. Therapeutic FMT Restores Normobiotic Microbial Ecologies

Next, to evaluate if the resolution of inflammation in FMT-treated mice was associated with changes in the gut microbiota composition, fecal samples of untreated, colitic, and FMT-treated mice were analyzed by 16S rRNA sequencing on the Illumina Miseq platform (Figure 6).

Principal component analysis (PCA) of unweighted UniFrac metric showed separation among treatment groups, with a significant difference between cDSS and cDSS + FMT (PERMANOVA, padj < 0.05) (Figure 6A; Appendix A).

As previously shown [45], chronic DSS treatment is associated with a decrease in alpha diversity upon chronic DSS treatment, particularly in species diversity indices (ANOVA; Simpson, *p* < 0.05; Shannon, *p* < 0.05) (Appendix A). At the end of FMT treatment in chronically colitic mice the overall bacterial composition of fecal samples at phylum level was not significantly changed as compared to DSS-treated mice (Figure 6B). Consistent with previous reports [46], no change in abundance was observed among the most abundant phyla, including *Firmicutes* and *Bacteroidetes* (Figure 6B). Nonetheless, at lower taxonomic levels, FMT-treated mice showed partial restoration of *Bifidobacteriaceae* and *Erysipelotrichaceae* families (Figure 6B and Appendix A), which are immunomodulatory short chain fatty acids (SCFA)-producing taxa, previously shown to be associated with healthy gut ecosystems [19].

Differential abundance analysis at genus level showed significant differences between the cDSS and cDSS + FMT groups, with increased *Parabacteroides* and *Ruminoclostridium* in cDSS mice (Wald test, *p* < 0.01), and increased *Lachnospiraceae*, *Anaeroplasma*, *Ruminococcaceae UCG-0013*, and *Ruminococcaceae UCG-0014* in cDSS + FMT mice (Wald test, *p* < 0.05) (Figure 6C).

Among these differentially abundant taxa, *Parabacteroides, Anaeroplasma*, and *Ruminoclostridium* ranked as the most important to the microbial community, as measured by mean decrease in accuracy, suggesting that FMT treatment has more impact on these taxa in the gut microbiome environment in colitic mice (Figure 6D).

At the species level, FMT-treated mice showed a significant decrease in *Parabacteroides goldstenii* (Figure 6E), as well as a partial decrease in *Bacteroides acidifaciens*, *Escherichia-Shigella*, and *Blautia* genera, taxa overrepresented in murine models of intestinal inflammation and in IBD patients [7]. On the contrary, *Odoribacter spp.* and *Lactobacillus C30A*, two SCFA-producers present at low levels in DSS-treated mice [19] and in IBD patients [14], showed a slight increase upon FMT (Figure 6E).

Next, the microbiota profiles were used for functional prediction using PICRUSt. Among the differentially represented functions between the two groups, we observed an increase in tryptophan metabolism (an intermediate of the aryl hydrocarbon receptor pathway), impaired in the context of intestinal inflammation [47], as well as in metabolism of amino acids associated with bacterial fermentation [48] in FMT-treated compared to cDSS mice.

Several pathways associated with bacterial hyperproliferation (phenylalanine metabolism, bisphenol degradation, transcription-related proteins, fatty acid biosynthesis) and inflammatory responses were reduced in FMT-treated mice. Among them, the metabolic pathway converting linoleic acid into arachidonic acid, which is highly active in UC patients [49], was significantly reduced upon FMT (Figure 6F).

Taken together, these findings suggest that the beneficial effects of FMT during chronic intestinal inflammation may originate from the functional reshuffling of the entire microbiota population towards anti-inflammatory activities.

## 4. Discussion

A growing interest is emerging for FMT as a therapy for infectious and chronic diseases characterized by microbial alterations. FMT has demonstrated to be highly effective for the treatment of antibiotic-resistant recurrent *Clostridium Difficile* infection (CDI) [22,23] and has shown promising results for treatment of Inflammatory Bowel Diseases (IBD), particularly ulcerative colitis (UC) [24,25,26].

Successful trials involving UC patients demonstrated that restoration of normobiosis correlates with clinical remission [24,25,26]. From a mechanistic point of view, we have recently shown that therapeutic FMT treatment in experimental models of acute colitis not only introduced beneficial modifications in the dysbiotic microbiota, but also impacted on frequencies and cytokine profiles of both innate and adaptive immune cells, as a consequence of an IL10-dependent amelioration of the inflammatory status [19].

Here we extended these observations to a chronic model of intestinal inflammation, which more closely resembles the clinical and immunological features observed in UC patients [11]. One important point that emerged from the human UC and CDI trials indicated that induction and maintenance of disease remission required multiple FMT administrations [22,23]. Based on these observations, to ensure proper engraftment of the healthy microbial communities, we adopted a protocol of repeated administration of healthy mucus and feces for 4 days per DSS cycle, and this protocol confirmed successful therapeutic outcomes. 

Our results are in agreement with the findings of the patient’s clinical trials, as well as largely with our previous observations in the acute experimental models, indicating conserved mechanisms linking gut microbiota and immune system functions. 

The successful RCTs in UC patients [24,25,26] demonstrated that clinical remission correlates with stable changes of the gut microbiota composition and metabolism towards functional normobiosis restoration [50]. Relevant changes in the levels of *Erysipelotrichaceae* and *Lactobacillaceae* have been described both in IBD patients and in animal models of intestinal inflammation [3,7,51]. Here we found that these taxa were also similarly altered in chronically inflamed mice, and their abundances were restored upon therapeutic FMT. Moreover, similarly to the core ecology capable to control intestinal inflammation in the acute DSS model [19], also in the chronic setting the presence of *Lactobacillaceae*, *Erysipelotrichaceae*, *Ruminococcaceae*, *Rikenellaceae*, and *Lachnospiraceae* families correlated with resolution of the intestinal inflammation. These taxa share similar metabolic functions (i.e, short chain fatty acid production, pH control, free radical scavenging) and are capable of controlling the overgrowth of pathogens or pathobionts and to support intestinal homeostasis. These metabolic pathways have been shown to be associated with the successful induction of remission in UC patients receiving FMT [50].

Here, the expansion of these beneficial taxa in FMT-treated mice, and the contraction of pro-inflammatory taxa, such as *Proteobacteria*, *Enterobacteriaceae*, *Enterococcaceae*, *Christensenellaceae*, and *Streptococacceae* [7,14], also suggested a shift in the metabolic functions of the microbial ecology. Indeed, Phylogenetic Investigation of Communities by Reconstruction of Unobserved States (PICRUSt) analyses predicted that the restoration of a normobiotic ecology is associated with increased tryptophan (Trp) and isoleucine/leucine metabolism. 

The importance of Trp metabolism in intestinal homeostasis maintenance is underlined by the observation that IBD patients manifest low serum levels of Trp [52] and Trp-deficient mice show aggravation of colitis and reduced levels of antimicrobial peptides [53]. Re-equilibration of Trp levels, for example by dietary intervention, is sufficient to exert anti-inflammatory effects in experimental colitis models, to decrease colonic IFNγ and TNF, and to decrease the expression of pro-apoptotic factors [54], findings that are congruent with the results we showed here. Moreover, the rate-limiting enzyme for Trp metabolism, IDO1, is also expressed by epithelial and antigen presenting cells and its regulation modulates immune responses both locally and systemically. Finally, the gut microbiota metabolizes Trp to indole derivatives, which act as ligands for the aryl hydrocarbon receptor (AhR) receptor, inducing local production of IL22 and contributing to tissue repair and homeostasis maintenance [55], as suggested by the restored barrier functions in FMT-treated mice. 

Interestingly, the increase in leucine and isoleucine metabolism suggests higher nitrogen source availability for fermentation, which is one of the key metabolic pathways employed by the enteric anaerobic microbiota to produce SCFA. These SCFA-producing fermenting bacteria, in turn, stimulate mucus and IgA production, promote tolerance via Treg induction, inhibit nuclear factor κB functions, enhance epithelial barrier integrity and repair, and promote competitive exclusion of pathogens [56]. 

On the other hand, PICRUSt prediction indicates a reduction of the linoleic acid metabolic pathway in FMT-treated mice. Linoleic acid, an n-6 polyunsaturated fatty acid, has been implicated in inflammatory processes in the gut [49]. It can be metabolized into arachidonic acid, whose metabolites, including prostaglandin E2, leukotriene B4, and thromboxane A2, have pro-inflammatory properties and are increased in the mucosa of patients with ulcerative colitis and in animal models of intestinal inflammation [57,58,59]. The concentrations of linoleic acid and of its metabolic products correlate with the degree of histological inflammation and medications currently used for IBD patients, such as 5-aminosalicylic acid, inhibit their formation [59]. Hence, the restoration of a normobiotic microbial ecology upon FMT treatment is additionally associated with changes in functional activities involved in tissue homeostasis.

The mucosal immune system is pivotal to initiate and sustain intestinal inflammation. T cells, in particular, exert pathogenic roles upon the recognition of bacterial antigens [10,60,61]. We [13] and others [62,63] reported that IBD patients show increased amounts of pathogenic intestinal Th1, Th17, and Th1/17 cells. Moreover, mucosal iNKT cells, a subset of lipid-specific T lymphocytes, can secrete pro-inflammatory cytokines in the gut [14] and affect bacterial colonization of the intestine [28,64]. Consistently, iNKT cells also perform pathogenic functions towards the intestinal epithelium in response to IBD-derived intestinal bacteria [14]. Here we showed that the functional shift in intestinal microbial ecology observed upon FMT treatment was sufficient to trigger a reduction in the expression of colonic inflammatory genes, such as *infg*, *il1b*, and *tnf* [19,27,65].

Furthermore, our data indicate that variations in the gut microbial ecology are capable of specifically controlling the proliferative capability of mucosal pathogenic T cell populations and to abolish their pro-inflammatory cytokine potential, both for adaptive and innate immune cell populations [66].

IL-10 is an anti-inflammatory cytokine known for critically contributing to intestinal immune homeostasis [67]. During acute colitis, IL-10 secretion by APC and T cells upon FMT was temporally linked to the resolution of inflammation [19]. Here, we similarly observed a restoration of IL-10 mucosal levels, confirming a pivotal function of IL-10 in intestinal homeostasis maintenance [68], also in the context of chronic intestinal inflammation. 

Finally, although not statistically significant, FMT treatment reduced the frequency of MHCII^+^ professional antigen-presenting cells in the colonic lamina propria, suggesting that bacterial-antigen presentation to CD4^+^T cells plays an important role in sustaining intestinal immune cell pathogenicity, also in the chronic inflammation context.

## 5. Conclusions

Here we demonstrate that the modulation of intestinal microbiota by FMT during chronic experimental colitis influences adaptive and innate mucosal immune responses. Normobiosis restoration, both in terms of microbial composition and metabolic functions, contributes to controlling immune-mediated intestinal inflammation. These findings provide key information to elucidate the complex interplay between the immune system and the gut microbial ecosystem during chronic inflammatory conditions, especially in light of the increasing implementation of FMT to treat diverse human pathologies.

## Figures and Tables

**Figure 1 cells-08-00517-f001:**
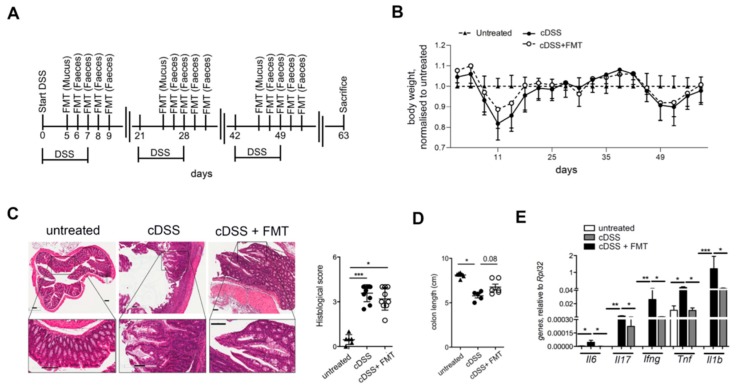
Therapeutic faecal microbiota transplantation (FMT) ameliorates Dextran Sodium Sulphate (DSS)-induced chronic inflammation. (**A**) Schematic representation of FMT treatment in the chronic DSS (cDSS) colitis model. (**B**,**C**) Weight loss (**B**) and hematoxylin-eosin (H&E) staining ((**C**), scalebar 100 µm, left) and cumulative histological score ((**C**), right) of colon specimens obtained from untreated (closed triangles), colitic (closed circles), or colitic mice treated with FMT (open circles). (**D**) Colon length of untreated (closed triangles), colitic (closed circles), or colitic mice treated with FMT (open circles). (**E**) Colonic expression levels of *Il6*, *Il17*, *Ifng*, *Tnf*, and *Il1b* in untreated (white bars), colitic (black bars), and FMT-treated (grey bars) mice. Statistical significance was calculated with unpaired two-tailed Mann-Whitney test (two-tailed) and expressed as mean ± SEM. Values of *p* < 0.05 (*), *p* < 0.01 (**), and *p* < 0.001 (***) were considered as statistically significant. Outliers were detected with Grubb’s test.

**Figure 2 cells-08-00517-f002:**
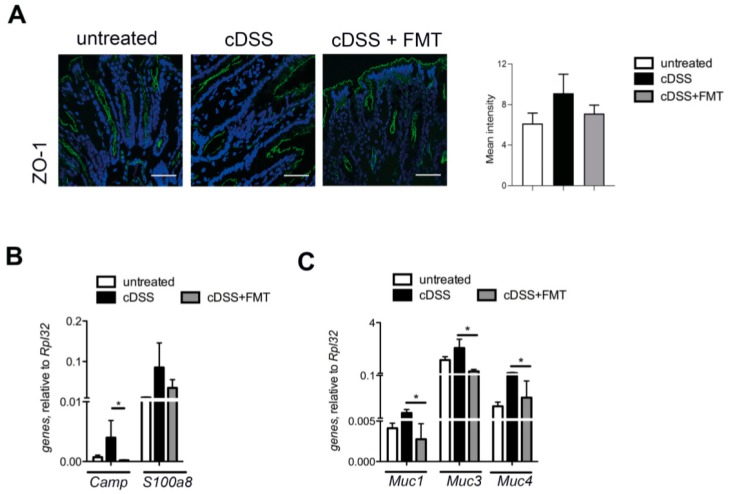
Gut intestinal barrier analysis in cDSS upon FMT. (**A**) ZO-1 immunofluorescence (IF) and quantification of untreated (left panel), cDSS (middle panel), and chronic DSS + faecal microbiota transplantation (cDSS+FMT) treated (right) mice colonic mucosa. Scalebar, 10 µm. (**B**) *Camp* and *S100a8* and (**C**) *Muc1*, *Muc3*, *Muc4* expression by colonic mucosa of untreated (white bars), cDSS treated (black bars), and cDSS+ FMT treated (grey bars) mice. Significance was determined using unpaired two-tailed Mann-Whitney test and expressed as mean ± SEM. Outliers were detected with Grubb’s test. *p* < 0.05 (*) were regarded as statistically significant.

**Figure 3 cells-08-00517-f003:**
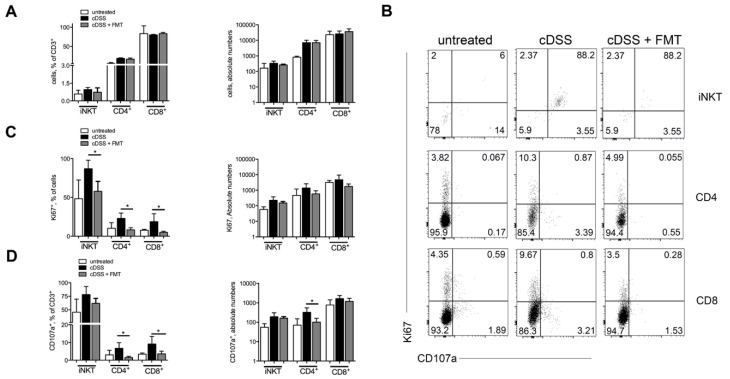
Therapeutic FMT modulates T cell phenotypes. (**A**) Frequencies and absolute numbers of colonic CD4^+^ T, CD8^+^ T cells, and Invariant Natural Killer T (iNKT) cells in untreated (white bars), chronic-DSS-treated (black bars), and FMT-treated (grey bars) mice 63 days after starting DSS administration. (**B**) Representative dot plots and (**C**,**D**) frequencies and absolute numbers of Ki67^+^ (**C**) and CD107a^+^ (**D**) colonic CD4^+^, CD8^+^ T cells, and iNKT cells in untreated (white bars), DSS-treated (black bars), and FMT-treated (grey bars) mice in the chronic setting. Significance was determined using unpaired Mann-Whitney test and expressed as mean ± SEM. *p* < 0.05 (*), *p* < 0.01 (**), *p* < 0.001 (***) were regarded as statistically significant. Total untreated *n* = 8, cDSS-treated *n* = 10, cDSS + FMT *n* = 10 in 2 independent experiments. Outliers were detected with Grubb’s test.

**Figure 4 cells-08-00517-f004:**
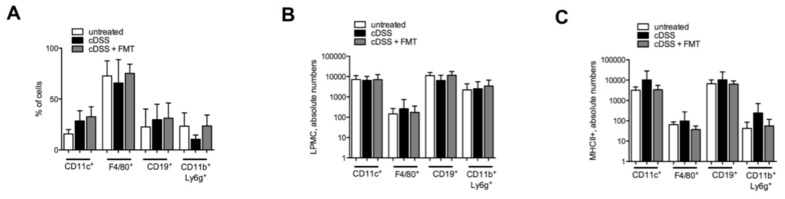
Antigen-presenting cells frequency and phenotype in cDSS-treated mice upon FMT. (**A**,**B**) Frequency (**A**) and absolute numbers (**B**) of colonic dendritic cells (CD45.2^+^CD3^−^CD11c^+^), macrophages (CD45.2^+^CD3^−^F4/80^+^), B cells (CD45.2^+^CD3^−^CD19^+^), and neutrophils (CD45.2^+^CD3^−^Ly6g^+^CD11b^+^) in cDSS-treated (black bars) and cDSS+FMT-treated (white bars) mice. (**C**) Absolute numbers of MHC-II expression in colonic dendritic cells, macrophages, B cells, neutrophils in cDSS-treated (black bars), and cDSS + FMT-treated (white bars) mice.

**Figure 5 cells-08-00517-f005:**
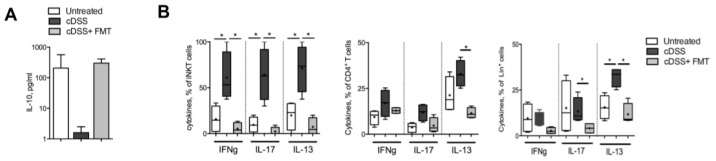
FMT reduces immune cell pro-inflammatory cytokine profile. (**A**) IL10 protein levels in colonic tissues of untreated (white bars), cDSS-treated (black bars), and cDSS+ FMT-treated (grey bars) mice. (**B**) IFNγ, IL17 and IL13 production by iNKT cells (left panels), CD4^+^ T cells (middle panels), and Lineage positive cells (CD11c, CD11b, CD19, right panels) in untreated (white boxes), cDSS-treated (Black boxes), and cDSS + FMT (Light grey boxes) mice. Significance was determined using unpaired Mann-Whitney test and expressed as mean ± SEM. *p* < 0.05 (*) were regarded as statistically significant. Outliers were detected with Grubb’s test.

**Figure 6 cells-08-00517-f006:**
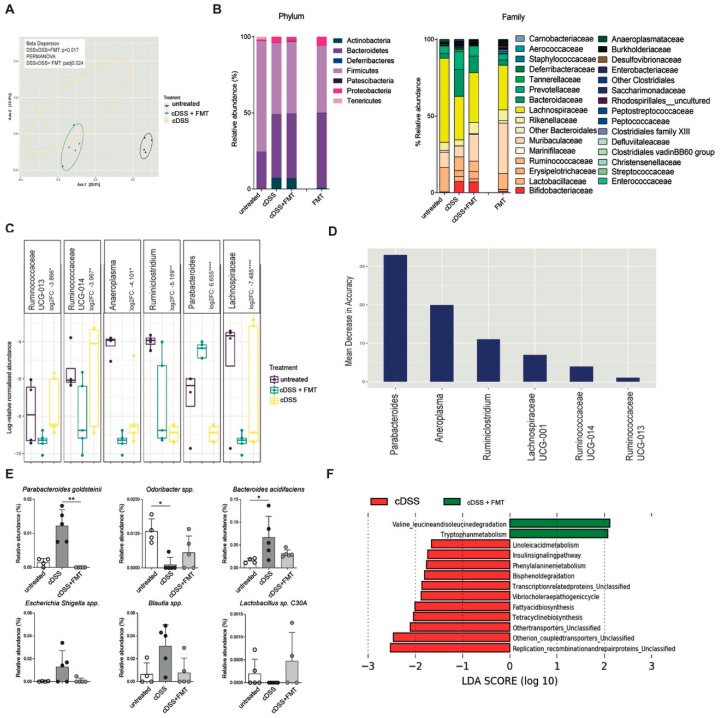
Gut microbiota analysis upon Faecal Microbiota Transplantation (FMT) in the chronic colitis model. (**A**) Microbiome clustering based on Principal Coordinate Analysis (PCoA) of unweighted UniFrac distances of fecal gut microbiota derived from untreated (violet dots), cDSS- (blue dots), and cDSS + FMT (yellow dots)-treated mice. Significant differences in beta-diversity between cDSS and cDSS + FMT groups were determined by beta-dispersion and Permutational Multivariate analysis of variance (PERMANOVA) with *p*-value correction by Bonferonni (padj), with *p* < 0.05 as cut-off for significance. (**B**) Bar plots showing taxonomic composition in untreated, cDSS, and DSS + FMT groups, including only taxa with relative abundance >1%, at phylum (left panel) and family (right panel) level. (**C**) Log-relative normalised abundances in untreated (violet boxes), cDSS- (blue boxes), and cDSS + FMT (yellow boxes)-treated mice of genera found to be differentially abundant in cDSS and cDSS + FMT mice, as determined by DESeq2. Log2 fold change (log2FC) relative to cDSS group is reported for each genus. Negative log2FC indicates decreased abundance in cDSS compared to cDSS + FMT, whereas positive log2FC indicates increased abundance in cDSS compared to cDSS + FMT. Significant differences in log2FC were determined by Wald test with *p*-value correction by Benjamin and Hochberg. *p* < 0.05 (*), *p* < 0.01 (**), *p* < 0.001 (***), *p* < 0.0001 (****) were regarded as statistically significant. (**D**) Ranking of differentially abundant taxa in cDSS and cDSS + FMT mice by importance to microbial community, as determined by their Mean Decrease in Accuracy (MDA) using a Random Forest classifier. Higher MDA means greater importance. (**E**) Relative abundances of bacterial species in untreated (white bars), cDSS (dark grey bars), and cDSS + FMT (light grey) treated mice. (**F**) Predicted metabolic attributes between cDSS and cDSS + FMT groups. Sequences were used to predict functions against the Kyoto Encyclopedia of Genes and Genomes (KEGG) database, which is implemented in Phylogenetic Investigation of Communities by Reconstruction of Unobserved States (PICRUSt) software package.

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
