# Peer review of "Fecal Microbiota Transplantation Controls Murine Chronic Intestinal Inflammation by Modulating Immune Cell Functions and Gut Microbiota Composition"

_cells, 2019, doi:10.3390/cells8060517_

Round 1

Reviewer 1 Report

Inflammatory bowel diseases (IBDs) display a severe health burden and are expected to rise in prevalence. Interestingly, stool transplantations from healthy donors to diseased individuals seem to be promising approaches because they affect the composition of the microbiota, however, underlying mechanisms are poorly understood. In their study, the authors aimed to elucidate the effect of faecal microbiota transplantation in a mouse model for chronic, DSS-associated colitis. For this purpose, they have used a comprehensive approach by analyzing untreated and DSS-treated animals as well as DSS-treated mice which underwent microbiota transplantation regarding their colitis burden, T cell phenotypes and gut permeability. In addition, they have performed 16S sequencing to analyze the composition of the microbiota. While their findings on this topic indicate the relevance of microbiota transplantation in colitis and will be of interest to the readership of Cells, the impact of the study is clearly limited due to the fact that in several experiments the untreated mice have not been analyzed. A number of major and minor points should be addressed prior to publication as detailed below.

To assess intestinal barrier function, IF staining for ZO-1 was performed in DSS-treated animals. Please perform ZO-1 staining on colons isolated from untreated animals and show a quantification of the staining intensity.

Please add the T cell characteristics of untreated mice to Figure 3.

Due to severe abdominal pain during DSS treatment, mice are usually treated with analgesics (for example buprenorphine). Were the mice treated with analgesics? Please add this information to the manuscript.

Was the presence of occult blood and the stool consistency checked in order to evaluate disease activity during the DSS treatment? If so, please add the results to the manuscript.

Since Figure 1B shows the body weight of mice during the complete treating period, please change the title of the y-axis to “Body weight” instead of “Starting weight”. Moreover, please show the body weight in gram or normalize to 1 instead of 100%. It does not make sense that the untreated animals have a body weight of 120% at the end of the experiment.

In the legend of Figure 1 the authors mention the n-numbers as follows: “Total cDSS-treated n=10, DSS+FMT n=10 in 2 independent experiments.” (line 244-245). How many untreated animals were analyzed? Please add the n-numbers to the Materials and Methods section.

Please add p-values to the bar graph in Figure 1C and show smaller magnifications of the colon section to give a better overview on the epithelial morphology.

There are confusions when referring to the figures in the text. For instance, when describing that “Nonetheless, T cell populations isolated from the colon of FMT treated mice showed a reduced proliferative capacity as compared to those isolated from colitic mice that had not received the FMT treatment (Fig1B,C).” (line 270-272), the authors probably meant Figure 3 B and C.

In the legend of Figure 2, a description for Figure 2A, B and C is given, however, the authors forgot to add a “C” to the figure.

Please add a paragraph on “Statistics” to the Methods section.

The style is not always consistent (e.g. “hours” [line 139] and “hrs” [line 154];“MUC1” in Figure 2B and “muc1” in the figure legend; in some graphs the mice are labeled as “untreated”, “cDSS” and “cDSS + FMT”, in others as “UT”, “DSS” and “DSS + FMT”).

Please use commas to separate groups of thousands and a period as a decimal separator (example in line 154: “Nuclei were counterstained with DAPI (1:30.000)”).

The manuscript has many typing and grammatical errors which need to be rectified.

The authors did not pay attention to the nomenclature. Human gene names should be written in capital letters and be italicized. Human proteins are written in capital letters. Murine gene names should be written in small letters (only the first letter is capitalized) and be italicized. Murine proteins are written in capital letters.

Please define meanings of acronyms the first time they are being used.

Author Response

We thank the reviewer for his/her appreciation of our work. Here below a point by point reply to the concerns raised.

·     To assess intestinal barrier function, IF staining for ZO-1 was performed in DSS-treated animals. Please perform ZO-1 staining on colons isolated from untreated animals and show a quantification of the staining intensity.

We performed the suggested IF experiments and quantified ZO-1 in untreated, DSS-treated and DSS+FMT treated mice. The results indicate that FMT-treatment in colitic mice restores ZO-1 levels to those of untreated animals. The novel experiments have been added in Revised Figure 2A

·       Please add the T cell characteristics of untreated mice to Figure 3

We added the missing T cell characteristics of untreated mice in the Revised Figure 3 and in the Revised Supplementary Figure 2

·      Due to severe abdominal pain during DSS treatment, mice are usually treated with analgesics (for example buprenorphine). Were the mice treated with analgesics? Please add this information to the manuscript

Within the protocol approved by Italy’s Ministry of Health (Auth 415/2017) and also according to the veterinarian of our Animal facility, we did not treat mice with analgesics. We added this information in the revised material and methods section.

·       Was the presence of occult blood and the stool consistency checked in order to evaluate disease activity during the DSS treatment? If so, please add the results to the manuscript.

We did not check for occult blood and/or stool consistency during DSS treatment.

·       Since Figure 1B shows the body weight of mice during the complete treating period, please change the title of the y-axis to “Body weight” instead of “Starting weight”. Moreover, please show the body weight in gram or normalize to 1 instead of 100%. It does not make sense that the untreated animals have a body weight of 120% at the end of the experiment

We thank the reviewer for the comments. We changed the title of the axis as suggested by the reviewer and changed the representation of the body weight variations according to his/her suggestions. The novel graph has been done by normalizing the body weight for each treatment measurement to that of untreated mice (Revised Figure 1 B)

We had chosen the previous representation of the body weight since it was already widely used in the literature (1) and we also recently used the same representation in our recent Nat Comm paper (REF #19 of the manuscript).

To explain the meaning of the “120%” of increase in the untreated mice we provided for the reviewer a table of the raw data (in grams) of Exp1 (Table 1 for Reviewer only). These data show that untreated mice during the 60 days of the experiment gain weight, thus explain the “120%” increase of weight as compared to their starting weight, set as 100%.

Table 1 for reviewer only

·      In the legend of Figure 1 the authors mention the n-numbers as follows: “Total cDSS-treated n=10, DSS+FMT n=10 in 2 independent experiments.” (line 244-245). How many

untreated animals were analyzed? Please add the n-numbers to the Materials and Methods section

We analysed n=8 untreated mice in the 2 independent experiments. This information has been added in the legend of Fig1 and in the material and methods section.

·      Please add p-values to the bar graph in Figure 1C and show smaller magnifications of the colon section to give a better overview on the epithelial morphology

We added the p-values in the Revised Figure 1C. We changed also the histological pictures of Figure 1C to better highlight the changes in the epithelial morphology of the colon sections, by adding an insert at a bigger magnification of the H&E staining for each experimental group below each panel.

·      There are confusions when referring to the figures in the text. For instance, when describing that “Nonetheless, T cell populations isolated from the colon of FMT treated mice showed a reduced proliferative capacity as compared to those isolated from colitic mice that had not received the FMT treatment (Fig1B,C).” (line 270-272), the authors probably meant Figure 3 B and C

We apologize with the reviewer for the confusion. We corrected the text accordingly

·      In the legend of Figure 2, a description for Figure 2A, B and C is given, however, the authors forgot to add a “C” to the figure

We apologize with the reviewer and added the missing label to C panel in the Revised figure 2.

·      Please add a paragraph on “Statistics” to the Methods section

We added the paragraph to the material and methods section.

·      The style is not always consistent (e.g. “hours” [line 139] and “hrs” [line 154];“MUC1” in Figure 2B and “muc1” in the figure legend; in some graphs the mice are labeled as “untreated”, “cDSS” and “cDSS + FMT”, in others as “UT”, “DSS” and “DSS + FMT”

We uniformed the style in the text, in the figure legend and in the figures.

·       Please use commas to separate groups of thousands and a period as a decimal separator (example in line 154: “Nuclei were counterstained with DAPI (1:30.000)

Done.

·      The manuscript has many typing and grammatical errors which need to be rectified.

We apologize.The manuscript has been carefully checked and typos and grammatical errors have been corrected.

·      The authors did not pay attention to the nomenclature. Human gene names should be written in capital letters and be italicized. Human proteins are written in capital letters. Murine gene names should be written in small letters (only the first letter is capitalized) and be italicized. Murine proteins are written in capital letters

We corrected the nomenclature of murine genes and proteins accordingly.

·      Please define meanings of acronyms the first time they are being used.

Done.

1.     Kiesler, Fuss Strober

Experimental models of inflammatory bowel disease

Cell mol Gastroenterol Hepatol 2015 mar 1; 1(2) 154-170

Reviewer 2 Report

1)      Authors describe that mice were divided by gender. It is not clear whether results are from males, females or whether they are from both groups.  According to our observations males are more sensitive to the influence of DSS and we cannot compare groups of different sexes.

2)      Microbiota plays crucial role in the severity of gut inflammation induced by DSS treatment. Did you analyze microbial composition of feces of mice to whom was done the transfer (FMT)? 

3)      It is known that the composition of the diet affects the development of colitis.

It is necessary to describe the composition of the diet.

4)      DSS induces inflammation in colon (model of ulcerative colitis (UC)). Why did you not analyze  ileal part of the intestine too?

5)      Did you apply the acute model of DSS or another model of UC ?

6)      Fig 1c is very small, the text in Fig 6 is very small and therefore unreadable.

Author Response

·        Authors describe that mice were divided by gender. It is not clear whether results are from males, females or whether they are from both groups.  According to our observations males are more sensitive to the influence of DSS and we cannot compare groups of different sexes

We thank the reviewer for to possibility to clarify this point. Based on our similar experience, we had performed our experiments always only with females. We added this information in the revised material and methods section.

·        Microbiota plays crucial role in the severity of gut inflammation induced by DSS treatment. Did you analyze microbial composition of feces of mice to whom was done the transfer (FMT)? 

We did not perform dedicated 16S sequencing to recipient mice before FMT. Nonetheless, the mice used for the experiments are genetically identical, littermates of the same age and sex and are housed in the same room in the animal facility, thus harboring the same gut microbiota. At the beginning of the experiment untreated (recipient) mice are reshuffled in the different experimental groups. For this reason it is reasonable to believe that at the beginning of the experiment mice receiving DSS or DSS + FMT treatment will have the same gut microbiota of untreated mice. We can therefore consider the gut microbiota of untreated mice homogenous to that of DSS and DSS + FMTtreated mice before the treatments.

·        It is known that the composition of the diet affects the development of colitis. It is necessary to describe the composition of the diet

The standard autoclaved diet provided to mice in our Animal facility is the VRF1 (SDS, UK). It is a certificated rat and mouse breeding diet also suitable for maintenance. Contains elevated levels of certain heat labile vitamins which make it suitable for autoclaving and for animals with high vitamin requirements (i.e. SPF / Germ free). The diet is coated with inert silicon dioxide to reduce the potential for clumping during autoclaving. Designed to be fed ad-libitum with fresh water.

Ingredients: Wheat, Dehulled Extracted Toasted Soya,Wheatfeed, Barley, Dehulled Cooked Soya, Soya Oil, Calcium Carbonate, Dicalcium Phosphate, Salt (NaCl), Silicon Dioxide, DL-Methionine, Choline Chloride, Ferrous Sulphate, Inositol, Magnesium Oxide, Zinc Oxide, L-Lysine, Manganese Oxide,Vitamin E (Tocopherol) Supplement, Vitamin B12 (Cyanocobalamin) Supplement,Vitamin B1 (Thiamine), Nicotinic Acid,Vitamin A (Retinol) Supplement, Copper Sulphate, Calcium-D-Pantothenate,Vitamin B6 (Pyridoxine),Vitamin K3 (Menadione), Selenium 1% Supplement,Vitamin B2 (Riboflavin), Biotin-Supplement, Folic Acid, Calcium Iodate,Vitamin D3 (Cholecalciferol) Supplement.

·        DSS induces inflammation in colon (model of ulcerative colitis (UC)). Why did you not analyze ileal part of the intestine too?

Based on our previous experience and on the data present in literature (1,2,3,4), the chronic DSS colitis model affects mostly, and is limited to, the distal colon. For this reason we did not analyse the ileal part of the intestine.

·        Did you apply the acute model of DSS or another model of UC?

Yes, we performed similar experiments in the acute DSS model of colitis and found that also in that condition FMT results protective. We published the data in Burrello, C., et al., “Therapeutic faecal microbiota transplantation controls intestinal inflammation through IL10 secretion by immune cells”. Nat Commun, 2018. 9(1): p. 5184.

·        Fig 1c is very small, the text in Fig 6 is very small and therefore unreadable

We increased the text in the Revised Figure 6 to render it more readable.

1.     Kiesler, Fuss Strober

Experimental models of inflammatory bowel disease

Cell mol Gastroenterol Hepatol 2015 mar 1; 1(2) 154-170

2.     Ulrike Erben, Christoph Loddenkemper, Katja Doerfel, Simone Spieckermann, Dirk Haller, Markus M Heimesaat , Martin Zeitz, Britta Siegmund, Anja A Kühl

A guide to histomorphological evaluation of intestinal inflammation in mouse models

Int J Clin Exp Pathol 2014;7(8):4557-4576

3.     Okayasu I., Hatakeyama S., Yamada M, Ohkusa T, Inagaki Y, Nakaya R

A novel method in the induction of reliable experimental acute and chronic ulcerative colitis in mice

Gastroenterology, Volume 98, Issue 3, March 1990, Pages 694-702

4.     Cooper HS, Murthy SN, Shah RS, Sedergran DJ

Clinicopathologic study of dextran sulfate sodium experimental murine colitis

Lab Invest 1993, Aug 69(2) 238-49

Round 2

Reviewer 1 Report

According to the authors' letter and the manuscript, all concerns and suggestions have been addressed. 

However, there seems to be a problem with the figures. 

For instance, the authors wrote: "We performed the suggested IF experiments and quantified ZO-1 in untreated, DSS-treated and DSS+FMT treated mice. The results indicate that FMT-treatment in colitic mice restores ZO-1 levels to those of untreated animals. The novel experiments have been added in Revised Figure 2A."

When looking at the manuscript, the legend of Figure 2 refers to IF of untreated and treated mice (plus quantification) but the according figures cannot be found in the manuscript. Moreover, in the legend of Figure 2, figures A, B, C are described but the Figure 2 only contains A + B.

Thus, it seems like the figures have not been updated in the revised manuscript.

Author Response

We thank the reviewer for his/her appreciation.

We had uploaded both the revised manuscript with the corrected figure legend (which I copy here below) and the corrected figures (which I uploaded as .pdf and re-upload now):

Figure 2 Gut intestinal barrier analysis in cDSS upon FMT. (A) ZO-1 IF and quantification of untreated (left panel), cDSS (middle panel) and cDSS+FMT treated (right) mice colonic mucosa. Scalebar, 10 µm. (B) Camp and S100a8 and (C) Muc1, Muc3, Muc4 expression by colonic mucosa of untreated (white bars), cDSS treated (black bars) and cDSS+ FMT treated (grey bars) mice. Significance was determined using unpaired two-tailed Mann-Whitney test and expressed as mean±SEM. Outliers were detected with Grubb’s test. P < 0.05 (*) were regarded as statistically significant.

We hope that the corrections we made are now visible.
